# Allogeneic Limbal Transplants Integrate into the Corneal Surface and Lead to an Improved Visual Acuity

**DOI:** 10.3390/jcm12020645

**Published:** 2023-01-13

**Authors:** Christiane Kesper, Joana Heinzelmann, Anja Viestenz, Thomas Hammer, Sabine Foja, Marlene Stein, Arne Viestenz

**Affiliations:** Department of Ophthalmology, University Hospital Halle (Saale), 06120 Halle, Germany

**Keywords:** limbal stem cell deficiency, allogeneic limbal transplantation, keratoplasty, chemical burn, limbal stem cells, limbus

## Abstract

Limbal stem cell deficiency (LSCD) severely impairs vision and can lead to blindness. LSCD causes include chemical burns, infections, multiple previous operations and congenital malformations. Allogeneic limbal transplantation is a procedure for treating LSCD where prepared limbal tissue is attached using a double running suture during allogeneic penetrating keratoplasty (PKP). A total of 22 patients underwent ALT surgery between February 2019 and June 2022 at the University Hospital Halle (Saale). Regular follow-up was performed postoperatively every three months and included visual acuity testing, pressure measurement, slit lamp microscopic examination, fundoscopy, corneal topography and anterior segment optical coherence tomography (AS-OCT). The mean patient age was 69.5 years, and the mean follow-up was 19 months. All included patients had LSCD and multiple previous surgeries. Patient LSCD etiology was 59% infectious and 41% traumatic. ALTs integrated into corneal surfaces in all patients, demonstrated on AS-OCT. Since most patients initially received allogeneic limbal transplants, none of the operated eyes had surgical complications. Overall, visual acuity improved postoperatively from an initial 2.06 to 1.44 logarithm of the minimum angle of resolution (logMAR). Allogeneic limbal transplantation can be used to treat LSCD and its integration into the surrounding corneal tissue can be observed on AS-OCT.

## 1. Introduction

Corneal epithelial stem cells are located at the limbus and are necessary for the continuous renewal of the corneal epithelium. Damage to these cells leads to limbal stem cell deficiency (LSCD), which accompanies a massive loss of visual acuity, conjunctivalization of the cornea, and recurrent defects of the corneal surface. In many cases, the patients also suffer from blindness and pain. LSCD is often caused by ocular trauma, especially chemical burns, infectious diseases or congenital disorders. LSCD can occur in different forms, partial and complete.

For partial LSCD, there is locally limited destruction of the limbal epithelial stem cells, whereas in complete LSCD, all stem cells are damaged. This results in complete conjunctivalization and vascularization of the corneal surface in the form of a pannus. Due to this, the barrier function of the stem cells is lost, and the regeneration of the corneal epithelium by limbal epithelial stem cells is no longer guaranteed [1,2].

In 2022, a new therapeutic option for patients with LSCD with allogeneic limbal transplantation (ALT) was published. This procedure is carried out as part of penetrating keratoplasty (PK) [3].

The aim of this study was to obtain first long-term results of the allogeneic limbal transplantation and to compare them with the already established methods using the existing literature. Furthermore, the integration of allogeneic limbal grafts into the corneal surface should be demonstrated by AS-OCT.

## 2. Materials and Methods

### 2.1. Patients and Inclusion Criteria

A total of 22 eyes from 22 patients were included in the study. These patients corresponded to those in the work of Viestenz et al. and were supplemented by eight additional patients in whom surgery was performed [3]. The inclusion criteria were the clinical presence of LSCD and multiple previous operations.

### 2.2. Surgery

All patients underwent the surgery in the inpatient setting. It was carried out under general anesthesia. After a complete excision of the corneal pannus was necessary, a perforating keratoplasty was performed using a Hessburg–Barron trephine system. In some cases, single sutures were used for the fixation, whereas in other cases double running sutures in combination with single sutures were used. Afterwards, preparation of the limbal tissue started. One to eight limbal parts out of the sclerocorneal donor tissue were prepared. The number of limbal pieces was dependent on the diameter of the corneal graft, the LSCD severity and the size of the limbal pieces. These tissue parts were placed under the sutures where the strongest vascularization in the corneal tissue had been identified. Finally, a contact lens was placed to protect the limbal grafts.

### 2.3. Clinical Examination

Preoperatively, a best-corrected visual acuity (BCVA) test, complete ophthalmologic examination with a slit lamp, fundoscopy or ultrasound examination, photo-documentation, anterior segment optical coherence tomography (AS-OCT) and, if possible, video-keratography were performed. These examinations were repeated every 2–3 months as a part of the follow-up. The preoperative visual acuity and the visual acuity at the time of the last visit were used for the evaluation of visual acuity. This could be completed for all patients.

### 2.4. Monitoring of Integration of the Limbal Transplants

As a part of the AS-OCT, a thickness measurement of the limbal tissue grafts was performed with SPECTRALIS^®^ OCT. For this purpose, one of the transplants was selected as an example for each patient. This should be easily recognized on all performed measurements. In the first postoperative measurement, the position for the subsequent measurement times was then determined on this graft so that the evaluation could always take place at the same position on the graft over time. This was conducted manually with the Heidelberg software.

### 2.5. Statistic

Statistical analysis of the visual acuity results was performed using the Wilcoxon-Test in the absence of a normal distribution, which was tested by Anderson–DarlingTest. From the results of the Wilcoxon test, the *p*-value could be determined via the T-distribution. The calculations were performed using Excel.

## 3. Results

### 3.1. Epidemiological Characteristics

Twenty-two patients were included in the study between February 2019 and June 2022 (seven females, fifteen males). Ten of the affected eyes were left eyes, and twelve were right eyes. The mean age of the patients was 69.5 years (47–90 years). Nine of the patients (41%) developed LSCD due to chemical burns. Thirteen of the patients (59%) had LSCD because of an infection. Out of these patients, the most frequent cause of LSCD was herpetic infection. The median follow-up was 18.9 months.

All patients suffered from low visual acuity, recurrent corneal erosions and vascularization and had undergone multiple previous surgeries. Fourteen of the patients had received keratoplastic surgery several times previously.

### 3.2. Visual Acuity

Overall, 77.2% of patients showed an increase in visual acuity. The median average preoperative visual acuity (logMAR) was 2.06 ± 0.54, and it improved postoperatively to 1.44 ± 0.88 with a significance of 0.017. The results are shown in Figure 1. Differences were shown between patients with traumatic vs. infectious LSCD etiologies. Thus, the post-traumatic group showed a significant increase in visual acuity postoperatively (*p*-value = 0.017), whereas the increase in the postinfectious group was not significant (*p*-value = 0.15). In total, 88.8% of the trauma group and 58.3% of the infection group showed an increase in visual acuity. The results are shown in Table 1.

### 3.3. Thickness in AC-OCT

In total, the thickness measurements in the AS-OCT could be carried out in 13 of the patients. In the other patients, this was not possible due to patient noncompliance or poor image quality.

The initial thickness of the limbal portions differed from each other because no exactly equal limbal portions could be prepared during the preparation procedure.

There was a reduction in the thickness of limbal grafts in all patients. The individual courses are shown in Figure 2. The first measured ALT thickness averaged 247.3 ± 115.9 μm. Overall, by the time of the last measurement, there was an average reduction to 98.8 ± 52.0 μm, representing a mean reduction to an average of 42.9% of baseline. Limbal grafts were not dislodged in any patients.

An exemplary clinical case (P6) is shown in Figure 3.

### 3.4. Complications

A total of eight patients developed complications after the surgery (two patients out of the trauma group). Six patients out of the postinfectious LSCD group developed the following postoperative complications: five with a recurrence of corneal ulcer, which made a new PKP necessary in two eyes, and one eye had to be enucleated because of severe infectious keratitis with endophthalmitis.

The trauma patients were mostly free of complications; only one recurrent corneal erosion required treatment with autologous serum eye drops. One eye developed a raised intraocular pressure.

Overall, there were no complications in any of the operated eyes due to the initial prominence of the allogeneic limbal transplants, and none of the patients showed a relapse of LSCD.

## 4. Discussion

There are several approved therapeutic options for treating LSCD, for example, KLAL (keratolimbal allograft), CLET (cultivated limbal epithelial transplantation) and SLET (simple limbal epithelial transplantation).

Tsai and Tseng reported a limbal allograft transplantation in 1994, in which the allogeneic limbal tissue was transplanted to the ocular surface after pannectomy [4].

CLET was developed by Pellegrini et al. in 1997, whereby autologous or allogeneic limbal tissue was obtained, which was subsequently expanded ex vivo and later transplanted onto the eye in a second step [5]. This procedure was approved as Holoclar^®^ by the European Medicines Agency as a stem cell-based drug in the EU in 2015 [6].

In the SLET technique, autologous limbal tissue is harvested and cultured on an amniotic membrane in vivo [7].

In contrast to these two procedures, ALT does not pose a risk for the development of iatrogenic LSCD, since no autologous tissue is needed because allogeneic limbal tissue could be used, which is needed to perform PKP in the same patient. Because of this, the procedure is well-accepted by patients, as these patients are often very reluctant to undergo surgery on their healthy eye, even if it is “only” a small biopsy. The initial results on ALT [3] were confirmed in this work, showing graft survival and visual acuity increase over a longer period of time than in the previous study, averaging 18.9 months.

Another advantage of allogeneic limbal transplantation is that it can also be used in bilateral LSCD because, as described above, no autologous limbal tissue is needed. Due to the PKP performed at the same time as the ALT, a faster increase in visual acuity is also possible. After CLET/Holoclar^®^ or SLET, a subsequent PKP is often necessary, whereas with the ALT procedure, no two-stage procedure is needed.

In contrast to CLET/Holoclar^®^, a specialized laboratory is not needed, since no ex vivo cultivation is performed with ALT [3].

Similar to other techniques, ALT also results in a longer-term increase in visual acuity in complicated eyes.

In the study of Tsai and Tseng, a limbal allograft transplantation was performed and visual acuity was increased in 81.3% of patients with a follow-up of 18.5 months, which is comparable to our study [4].

Miri et al. demonstrated an 82% success rate in patients with LSCD who had received limbal stem cell transplantation, which is slightly higher than the success rate in our study. A total of 27 eyes were included, of which 12 were autologous and 15 allogeneic limbal grafts [8]. However, it should be noted that the mean preoperative visual acuity of 0.9 LogMAR was significantly better compared to that in our study (2.06). This suggests that the patients studied by Miri et al. had a less severe form of LSCD.

The long-term results of CLET were studied by Rama et al. [9]. This showed a success rate of 76% in a study of 112 patients, which is consistent with our results and the results of SLET. However, 46% of the patients required a second surgical procedure before the final best visual acuity could be achieved.

A study by Basu et al. showed a 76% success rate and a 75.2% increase in visual acuity at 1.5 years follow-up after autologous SLET surgery. The proportion of patients with a visual acuity increase is similar to that in our work. However, considering that Basu et al. included mainly patients with post-traumatic LSCD in the study and comparing this with the visual acuity increase in the trauma group in our work, the proportion of patients with visual acuity improvement after allogeneic limbal transplantation seems to be larger at 88.8%, but it should be added that in our study a much smaller number of patients was examined. In addition, Basu et al. showed that concurrent PKP was associated with a worse outcome and they postulated that graft survival was higher after autologous transplantation than after allogeneic transplantation. This could not be proven in our case, since none of the patients had a recurrence of LSCD. In addition, the efficacy of allogeneic SLET has also already been demonstrated. Here, the success rate was 83%, which corresponds to the results of our study. Furthermore, it could be shown that there was no difference between cadaveric tissue and tissue obtained from relatives [10,11].

However, in the literature comparison by Ganger et al., SLET and KLAL were superior to CLET in terms of functionality [11].

Additionally, ALT has a better outcome in patients with traumatic LSCD than in patients with infectious causes. This is in contrast to the results of Debourdeau et al., which showed a lower chance of success and an increased risk of rejection in patients with PKP due to trauma [12]. The previously described procedures of KLAL, CLET and SLET have been performed only in isolated cases in patients with infectious etiology of LSCD, making a comparison difficult here. The reason why the outcome is worse in patients with LSCD of infectious etiology remains as yet unknown. Perhaps the immune system is more active in patients with infectious LSCD, and thus problems are more likely to occur in the area of the graft. However, this remains only a conjecture.

In this study, the decrease in the thickness of ALT grafts over time was shown for the first time. This complements the previous results on this surgical method [3]. For this purpose, AS-OCT seems to be a suitable investigation. Here, for the first time, thinning of limbal grafts over time was demonstrated in the AS-OCT. Nevertheless, it remains unclear whether the thinning is bending due to tissue breakdown or integration into the surrounding tissue.

## 5. Limitations

Due to the very special patient clientele and the rarity of the findings, it has not been possible to create an appropriate comparison group. Therefore, a literature comparison had to be used for comparability. Another limitation is the feasibility of ASOCT in only 13 patients. This is a relatively high proportion of patients and can thus lead to a falsification of the results.

Since this study has so far only shown a mean follow-up of 18.9 months, no statement can yet be made on the long-term survival of the graft. The studies of Miri et al. and Basu et al. suggested that graft survival may be shortened in this type of transplantation. At present, however, it is not possible to make any statement about this in our patients.

## 6. Conclusions

Allogeneic limbal transplantation can be used to treat LSCD and its integration into the surrounding corneal tissue can be observed on AS-OCT. The results seem to be better in patients with a traumatic genesis of LSCD than in patients with an infectious genesis.

## Figures and Tables

**Figure 1 jcm-12-00645-f001:**
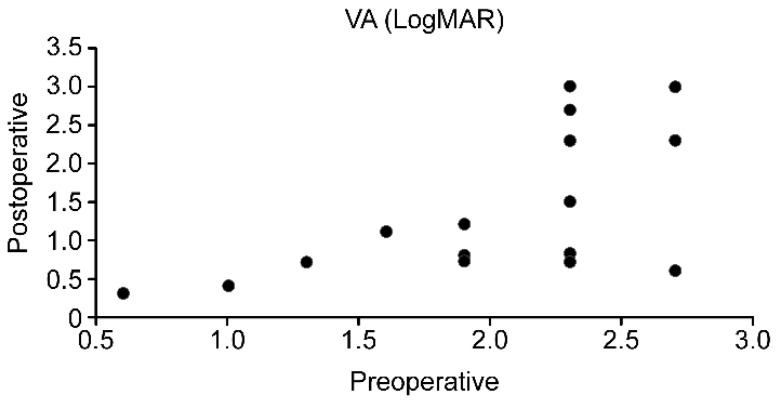
Comparison of visual acuity pre- and postoperatively.

**Figure 2 jcm-12-00645-f002:**
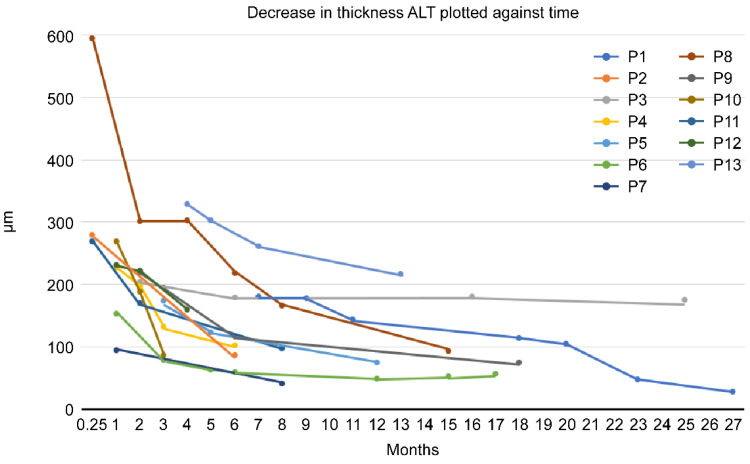
Decreased thickness of ALT plotted against time (P1–13 = numbered patients).

**Figure 3 jcm-12-00645-f003:**
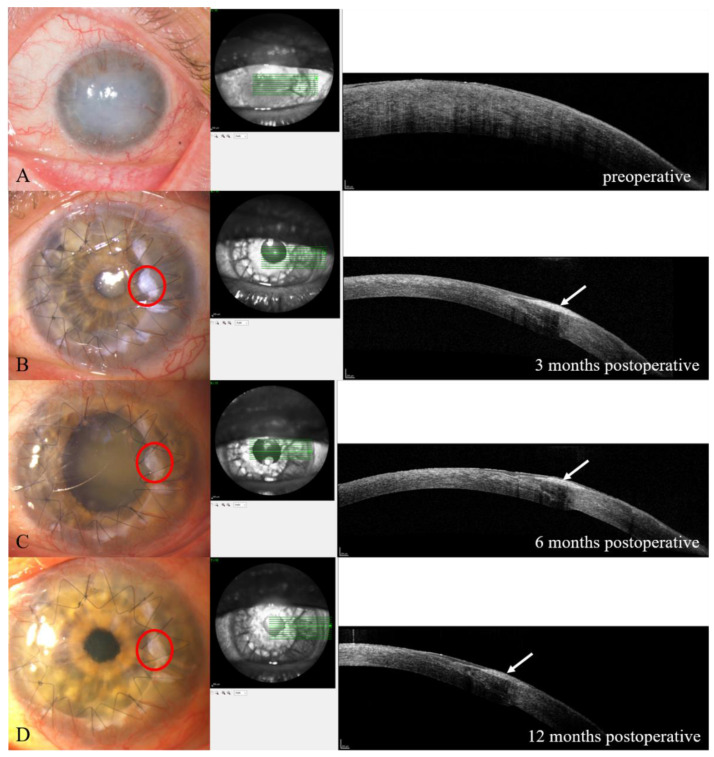
Clinical pictures of a patient with LSCD caused by ocular burn and the results of the AS-OCT. (**A**) = preoperative; (**B**) = 3 months postoperative; (**C**) = 6 months postoperative; (**D**) = 12 months postoperative; arrow = ALT in AS-OCT; red circle = ALT-transplant.

**Table 1 jcm-12-00645-t001:** Mean visual acuity of the traumatic and postinfectious groups.

	All Patients	Traumatic	Postinfectious
Preoperative mVA ± SD	2.06 ± 0.29	1.81 ± 0.49	2.24 ± 0.1
Postoperative mVA ± SD	1.44 ± 0.78	0.86 ± 0.51	1.87 ± 0.58
*p*-value, Percentage of patients with visual acuity increase	0.007, 77.2	0.017, 88.8	0.15, 58.3

mVA = mean visual acuity; SD = standard deviation.

## Data Availability

Data is contained within the article.

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
