# Peer review of "Allogeneic Limbal Transplants Integrate into the Corneal Surface and Lead to an Improved Visual Acuity"

_jcm, 2023, doi:10.3390/jcm12020645_

Round 1

Reviewer 1 Report

This work is well structured and well written in general lines. This article analyzes the results of a new technique described and published in August 2022 called by the same authors: Allogeneic limbal transplantation (ALT).

The first thing is that in the article where this technique was made known (Int Ophthalmol) it was published in August of this year with 14 cases and currently 20 are presented. I suppose that 6 more than when the technique was presented or I do not know if it is 20 new cases. Please clarify this. If there were only 6 cases added, specify the novelty of this article.

The ALT technique is original in introducing the fragments of cadaveric limbus in the continuous suture of penetrating keratoplasty, but the fact itself is not original, since the cadaveric corneal transplant together with sutured cadaveric limbus in different ways has already been described and has been for a long time (with the same name since 1994 Tsai RJ, Tseng SC. Human allograft limbal transplantation for corneal surface reconstruction. Cornea. 1994;13:389–400). For this reason, and because of the name given to it in the consensus published by Daya et al by the Cornea Society in 2011 in the Cornea journal (Daya SM, Chan CC, Holland EJ, Cornea Soc Ocular Surface P. Cornea Society Nomenclature for Ocular Surface Rehabilitative Procedures Cornea 2011;30(10):1115-9) already describes this technique as keratolimbal allograft or KLAL, I suggest changing the name of this technique to avoid confusion and thus make it more innovative.

Regarding the complications in the LSCD group caused by infections, why do the authors believe that there are so many important complications in this group and none in trauma one? This hypothesis should be reflected in the article.

In this sentence "In contrast to these two procedures, ALT does not pose a risk for the development of iatrogenic LSCD, since no allogeneic tissue is needed because allogeneic limbal tissue could be used, which is needed to perform PKP in the same patient." should be "since no autologous tissue".

Regarding the fact that of the 20 patients, only 13 could undergo AS-OCT, this should be considered as a limitation of the study because it is a high percentage and could be biased.

The advantages of allogeneic limbal transplantation are mentioned in this work, but their limitation is not mentioned, which is the viability of the transplant itself, since in longer series (not 19 months), Miri A et al saw that they reached the maximum to 5 years (Miri A, Al-Deiri B, Dua HS. Long-term Outcomes of Autolimbal and Allolimbal Transplants. Ophthalmology. 2010;117(6):1207-13). They should state it in the article.

Lastly, the authors should compare their results in their discussion with those obtained by other authors with allogenic limbal transplantation, without PKP.

Author Response

Dear Reviewer,

Thank you very much for your very helpful comments.

Point 1:

The first thing is that in the article where this technique was made known (Int Ophthalmol) it was published in August of this year with 14 cases and currently 20 are presented. I suppose that 6 more than when the technique was presented or I do not know if it is 20 new cases. Please clarify this. If there were only 6 cases added, specify the novelty of this article.

Answer 1:

You are absolutely correct regarding the number of patients. We have revised the manuscript again and added that the additional number is newly added patients.

Point 2:

The ALT technique is original in introducing the fragments of cadaveric limbus in the continuous suture of penetrating keratoplasty, but the fact itself is not original, since the cadaveric corneal transplant together with sutured cadaveric limbus in different ways has already been described and has been for a long time (with the same name since 1994 Tsai RJ, Tseng SC. Human allograft limbal transplantation for corneal surface reconstruction. Cornea. 1994;13:389–400). For this reason, and because of the name given to it in the consensus published by Daya et al by the Cornea Society in 2011 in the Cornea journal (Daya SM, Chan CC, Holland EJ, Cornea Soc Ocular Surface P. Cornea Society Nomenclature for Ocular Surface Rehabilitative Procedures Cornea 2011;30(10):1115-9) already describes this technique as keratolimbal allograft or KLAL, I suggest changing the name of this technique to avoid confusion and thus make it more innovative.

Answer 2:

We have taken out the abbreviation ALT and instead referred to allogeneic limbal transplantation.

Point 3:

Regarding the complications in the LSCD group caused by infections, why do the authors believe that there are so many important complications in this group and none in trauma one? This hypothesis should be reflected in the article.

Answer 3:

Why there were more complications in the infectious LSCD group than in the trauma group cannot be explained. So far, this remains only an observation that cannot (yet) be explained by us.

Point 4,5,6

In this sentence "In contrast to these two procedures, ALT does not pose a risk for the development of iatrogenic LSCD, since no allogeneic tissue is needed because allogeneic limbal tissue could be used, which is needed to perform PKP in the same patient." should be "since no autologous tissue".

Regarding the fact that of the 20 patients, only 13 could undergo AS-OCT, this should be considered as a limitation of the study because it is a high percentage and could be biased.

The advantages of allogeneic limbal transplantation are mentioned in this work, but their limitation is not mentioned, which is the viability of the transplant itself, since in longer series (not 19 months), Miri A et al saw that they reached the maximum to 5 years (Miri A, Al-Deiri B, Dua HS. Long-term Outcomes of Autolimbal and Allolimbal Transplants. Ophthalmology. 2010;117(6):1207-13). They should state it in the article.

Lastly, the authors should compare their results in their discussion with those obtained by other authors with allogenic limbal transplantation, without PKP.

Answer 4,5,6:

The error in the sentence formulation that you mentioned has been eliminated and the limitation that AS-OCT was not possible in all patients has been mentioned again in the point Limitations. Likewise, the limitation of the still missing 5-year results and the literature comparison to other comparable methods has been added.

I hope the changes are to your satisfaction. Once again, thank you very much for your comments.

I am looking forward to your new feedback.

Reviewer 2 Report

Summary

This article seems to provide knowledge about the integration of ALT tissue on the corneal surface. This ALT grafting performed after perforating keratoplasty is described in the introduction as an innovative therapy to LSCD. This seemingly niche knowledge may still be of interest to the scientific community in understanding this therapy described in another publication by the same team. The proposed results are rare because this ALT tissue graft is done during a perforating keratoplasty, therefore, few teams have to perform this type of therapy.

General concept Comments:
The purpose of this paper is not clear. From the title it seems that this paper is written to provide knowledge about the integration of ALT tissue on the corneal surface in LSCD patients, but no direct objective is written. In view of the results and the discussion, one wonders if this one is not finally intended to demonstrate the effectiveness of ALT. If this is the intention of the authors, it should be clearly stated.  
 The same applies to the abstract, where one wonders whether the aim is to show the integration of the tissue or to prove the effectiveness of ALTs.
A scientific article cannot be so evasive about its objectives because the methodology that follows cannot be judged properly. It is clear that it will not be the same protocol to demonstrate the efficacy of ALTs as to describe just observations on the integration of ALTs into the corneal tissue.
We can also criticise the simplicity of the article, which in the end comes to a very evasive conclusion and does not add much scientifically.

Specific Comments

Introduction
I strongly suggest that you mention the objective of your study.  
In the discussion section you mention an already known method for dealing with LSCDs, it might be a good idea to mention this in the introduction as well. What makes your therapy unique compared to others?

Methods
It would be welcome to describe the surgery. I understand that this has already been described previously (even though it is referenced in the Methods section) but a summary might make it easier to understand.  
Where do the transplanted tissues come from? These limbal cells are on a support or in a matrix. Where do they come from?  How are they obtained?
The statistics are not well described. To do a t-test did you check the normality of the distribution of the populations and the homoscedasticity (= equality of variances)? What software did you use?
For the thickness measurement, I advise you to describe your method a little more: number of measurements? What is the brand of your OCT? What software did you use?  

Results
3.2 "Differences were shown between patients with traumatic vs infectious LSCD etiologies. What is the statistical result (p-value)? What test did you do?

3.3 Why do patients have a different ALT thickness at D0? This is why the paragraph on the method should not be so short and not so detailed.

3.4 Here, if you wanted to show the efficacy of your ALT treatment or its superiority/equality to another treatment, you would have needed a control or comparison group: without ALT or with another therapy like Holoclar.

Discussion
In this section, you also seem to highlight the effectiveness of your treatment by comparing it to others. However, if we believe the title of your article, this is not what you want to describe. Here again we see the problem of forgetting to clarify the objective of an article.

Author Response

Dear Reviewer,

Thank you for your very helpful comments.

We have reviewed the article again and hope that the changes are to your satisfaction.

Point 1:

The purpose of this paper is not clear. From the title it seems that this paper is written to provide knowledge about the integration of ALT tissue on the corneal surface in LSCD patients, but no direct objective is written. In view of the results and the discussion, one wonders if this one is not finally intended to demonstrate the effectiveness of ALT. If this is the intention of the authors, it should be clearly stated. 

 The same applies to the abstract, where one wonders whether the aim is to show the integration of the tissue or to prove the effectiveness of ALTs.

A scientific article cannot be so evasive about its objectives because the methodology that follows cannot be judged properly. It is clear that it will not be the same protocol to demonstrate the efficacy of ALTs as to describe just observations on the integration of ALTs into the corneal tissue.

We can also criticise the simplicity of the article, which in the end comes to a very evasive conclusion and does not add much scientifically.

Answer 1:

The aim of this study has been revised again and formulated more precisely in the text. You are of course correct that emphasis has been placed on both visual acuity development and integration in AS-OCT. We have also considered this in the title of the manuscript.

Point 2:

Introduction

In the discussion section you mention an already known method for dealing with LSCDs, it might be a good idea to mention this in the introduction as well. What makes your therapy unique compared to others?

Answer 2:

Regarding your comment on the introduction, a presentation of already existing treatment options for LSCD has been added to the discussion part, as otherwise there would have been significant duplication in the introduction and the discussion. We hope that this is convenient for you.

Point 3:

It would be welcome to describe the surgery. I understand that this has already been described previously (even though it is referenced in the Methods section) but a summary might make it easier to understand. 

Where do the transplanted tissues come from? These limbal cells are on a support or in a matrix. Where do they come from?  How are they obtained?

The statistics are not well described. To do a t-test did you check the normality of the distribution of the populations and the homoscedasticity (= equality of variances)? What software did you use?

For the thickness measurement, I advise you to describe your method a little more: number of measurements? What is the brand of your OCT? What software did you use? 

"Differences were shown between patients with traumatic vs infectious LSCD etiologies. What is the statistical result (p-value)? What test did you do?

Why do patients have a different ALT thickness at D0? This is why the paragraph on the method should not be so short and not so detailed.

Answer 3:

The method section has also been revised and a more detailed description of the procedure of surgery as well as the measurement in AS-OCT has been added. Regarding the statistics, a re-examination was performed. Based on your very helpful comment, a correction of the statistics was made, this was described more clearly in the manuscript, and the results in the results section were adjusted.

Point 4 +5:

Here, if you wanted to show the efficacy of your ALT treatment or its superiority/equality to another treatment, you would have needed a control or comparison group: without ALT or with another therapy like Holoclar.

Discussion

In this section, you also seem to highlight the effectiveness of your treatment by comparing it to others. However, if we believe the title of your article, this is not what you want to describe. Here again we see the problem of forgetting to clarify the objective of an article.

Answer 4 +5:

Due to the special patient clientele, it was not possible for us to create an appropriate comparison group. Therefore, we have made a more detailed literature comparison with already existing results of other surgical methods and their results in the discussion.

We hope that the changes are to your satisfaction and again thank you for your comments.

I am looking forward to your new feedback.

Round 2

Reviewer 1 Report

Better.

Only in the second paragraph of point number 4, instead of putting "trabsplanted", write it correctly.

In addition, it should become clear at some point that despite the disadvantages of autotransplantation that have been clearly stated, it should be said that survival is higher in autotransplants than in allogeneic transplants.

Lastly, I don't understand the changes in statistical significance in Table 1. What results have changed to change that significance?

Author Response

Dear Reviewer,

Thank you again for your very helpful comments.

We have reviewed the article again and hope that the changes are to your satisfaction.

Point 1:

Only in the second paragraph of point number 4, instead of putting "trabsplanted", write it correctly.

Answer1:

Thank you for pointing this out. We have corrected the error.

Point 2:

In addition, it should become clear at some point that despite the disadvantages of autotransplantation that have been clearly stated, it should be said that survival is higher in autotransplants than in allogeneic transplants.

Answer 2:

We have added to the discussion on this.

Point 3:

Lastly, I don't understand the changes in statistical significance in Table 1. What results have changed to change that significance?

Answer 3:

Another reviewer had given us a note regarding the statistics. There was no normal distribution, which is why the statistical tests were changed and produced new results.

We hope that the changes are to your satisfaction and again thank you for your comments.

I am looking forward to your new feedback.

Reviewer 2 Report

Summary:

The aim of this article is to show the efficacy of allogeneic tissue transplantation therapy after transfixing keratoplasty in patients with limbal stem cell deficiency and at the same time to provide data on the integration of these tissues on the ocular surface. Despite the low novelty concerning the efficacy of this treatment which has already been published by the same team with an addition of 8 patients, the data on the integration of ALT tissues are new and rare given the exceptional context of their surgery (keratoplasty + LSCD). These results may provide new information to surgeons and researchers interested in allogeneic transplantation, ophthalmology and LSCD.

General concept Comments:

The authors highlighted the benefits (significant improvement in patients' visual acuity after surgery) and limitations of their study, allowing the reader to make up their own mind about the value of the therapy and the quality of the results.

However, some information is missing to make it clearer and the M&M part should be a bit more structured to make it more pleasant and clear for the reader.

Specific comments:

Thank you for all your replies and for the changes that have been made in your article.

The article seems to me much more comprehensible like this.

Your "Methods" section becomes "Materials & Methods".
It would be nicer to have titles in this M&M. For example: Inclusions; ALT grafting; Statistics; Monitoring ALT integration. This is just an example, and can be done as you prefer, but the reader would find it easier to find his way around with this construction.

It seems much more rigorous for the statistics.

Concerning visual acuity tests: did all the operated patients have these tests in pre-op and post-op? 22 patients pre-op and 22 post-op? If this is not the case, it is important to say so in order that the reader can form an opinion on the power of the statistical results given.

If I understand what you say in the "Method" section, you do your visual acuity measurements every 2-3 months after the operation. Is that correct? If so, the post op visual acuity results presented are obtained after how long? Don't you think this time would be important?

The discussion has become much more interesting. You show what your technique brings, you show the limits of your study. This transparency allows the reader to make up his or her own mind about the value of this therapy.

Please note that there is a typo in the Discussion section: "Tsai and Tseng reported a limbal allograft in 1994, in which allogeneic limbal tissue was trabnsplanted to the ocular surface after pannectomy [4].

A final remark, which remains an open question. Don't you think that the last sentence of your discussion should be a short conclusion that answers the purpose of your paper? A bit like you do in your abstract: "Allogeneic limbal transplantation can be used to treat LSCD and its integration into the surrounding corneal tissue can be observed on AS-OCT" but with the nuance about post-infectious results.

Author Response

Dear Reviewer,

Thank you again for your very helpful comments.

We have reviewed the article again and hope that the changes are to your satisfaction.

Point 1:

Your "Methods" section becomes "Materials & Methods".
It would be nicer to have titles in this M&M. For example: Inclusions; ALT grafting; Statistics; Monitoring ALT integration. This is just an example, and can be done as you prefer, but the reader would find it easier to find his way around with this construction.

Answer 1:

Thank you for the comment. We have renamed the section and divided it into subsections to make it easier to read.

Point 2:

Concerning visual acuity tests: did all the operated patients have these tests in pre-op and post-op? 22 patients pre-op and 22 post-op? If this is not the case, it is important to say so in order that the reader can form an opinion on the power of the statistical results given.

If I understand what you say in the "Method" section, you do your visual acuity measurements every 2-3 months after the operation. Is that correct? If so, the post op visual acuity results presented are obtained after how long? Don't you think this time would be important?

Answer 2:

Yes we were able to do the visual tests pre- and post-operatively on all 22 patients. We have included a reference to this in the M&M section. Visual acuity was checked every 2-3 months, but only the visual acuity at the time of the last visit (preliminary end of the follow up) was evaluated. This was also added in the M&M section.

Point 3:

Please note that there is a typo in the Discussion section: "Tsai and Tseng reported a limbal allograft in 1994, in which allogeneic limbal tissue was trabnsplanted to the ocular surface after pannectomy [4].

Answer 3:

Thank you for pointing this out. We have corrected the typing error.

Point 4:

A final remark, which remains an open question. Don't you think that the last sentence of your discussion should be a short conclusion that answers the purpose of your paper? A bit like you do in your abstract: "Allogeneic limbal transplantation can be used to treat LSCD and its integration into the surrounding corneal tissue can be observed on AS-OCT" but with the nuance about post-infectious results.

Answer 4:

Thank you for the good idea. We have added a subheading at the end of the publication where we briefly summarise this.

We hope that the changes are to your satisfaction and again thank you for your comments.

I am looking forward to your new feedback.